

# Subcellular compartmentalization of the plant antioxidant system: an integrated overview

Aleksandr Bobrovskikh[1,*], Ulyana Zubairova[1,2], Alexey Kolodkin[3,4] and Alexey Doroshkov[1,2,*]

[1] The Federal Research Center Institute of Cytology and Genetics, Siberian Branch of the Russian Academy of Sciences, Novosibirsk, Russian Federation
[2] Novosibirsk State University, Novosibirsk, Russian Federation
[3] University of Amsterdam, Amsterdam, Netherlands
[4] The University of Luxembourg, Luxembourg Centre for Systems Biomedicine, Luxembourg, Luxembourg
* These authors contributed equally to this work.

Corresponding author
Alexey Doroshkov, ad@bionet.nsc.ru

## ABSTRACT

The antioxidant system (AOS) maintains the optimal concentration of reactive oxygen species (ROS) in a cell and protects it against oxidative stress. In plants, the AOS consists of seven main classes of antioxidant enzymes, low-molecular antioxidants (e.g., ascorbate, glutathione, and their oxidized forms) and thioredoxin/ glutaredoxin systems which can serve as reducing agents for antioxidant enzymes. The number of genes encoding AOS enzymes varies between classes, and same class enzymes encoded by different gene copies may have different subcellular localizations, functional loads and modes of evolution. These facts hereafter reinforce the complex nature of AOS regulation and functioning. Further studies can describe new trends in the behavior and functioning of systems components, and provide new fundamental knowledge about systems regulation. The system is revealed to have a lot of interactions and interplay pathways between its components at the subcellular level (antioxidants, enzymes, ROS level, and hormonal and transcriptional regulation). These facts should be taken into account in further studies during the AOS modeling by describing the main pathways of generating and utilizing ROS, as well as the associated signaling processes and regulation of the system on cellular and organelle levels, which is a complicated and ambitious task. Another objective for studying the phenomenon of the AOS is related to the influence of cell dynamics and circadian rhythms on it. Therefore, the AOS requires an integrated and multi-level approach to study. We focused this review on the existing scientific background and experimental data used for the systems biology research of the plant AOS.

## INTRODUCTION

Aerobic metabolism provides great energetic benefits to organisms but has byproducts such as forms of reactive oxygen species (ROS), including singlet oxygen, superoxide

radical, hydrogen peroxide, hydroxyl radical. ROS plays a considerable role as signal molecules for cell processes (e.g., cell division, differentiation, immune response). However, excessive amounts of ROS are highly toxic and can damage cell membranes, oxidize various cellular substrates, and macromolecules. ROS initiates a cascade of reactions that enhance the production of hydroxyl ions and lipid peroxides in membranes (*Noctor & Foyer, 1998*). Furthermore, ROS triggers positive feedback by lipid oxidation, which can lead to damage to membranes. These processes can disrupt the transfer of electrons and lead to enhanced production of more ROS. The cumulative effect of excessive ROS can lead to disruption of the metabolism of individual cells, death of tissues, organs, and organisms, and cause so-called oxidative stress (*Gill & Tuteja, 2010*). In nonoptimal conditions, the production of ROS multiplies by many folds (*Orozco-Cardenas & Ryan, 1999*). So, for cellular metabolism, it is essential to maintain the optimal level of ROS.

Being one of the basic systems, the AOS arose in the time of aerobic photosynthetic metabolism appearance, which is approximately 2.4 billion years ago (*Hohmann-Marriott & Blankenship, 2011*). It is well known that the AOS is capable of controlling the concentration of ROS neutralizing their excess and protecting the plant cell against oxidative stress (*Ahmad et al., 2010*). For existing organisms, the AOS components could be encoded by several copies of the same enzyme class genes that may carry different functional loads and may be characterized by different modes of evolution. The genetic complexity of the AOS lies in its multicopy structure; for example, in the genome of *Arabidopsis thaliana* L., about 40 genes encode antioxidant enzymes (*Mittler et al., 2004*). Moreover, each enzymatic class in the AOS is represented in the plant genome in several copies, which, as a rule, have different localization in the cell compartments (cytosol, mitochondria, peroxisomes, chloroplasts).

The AOS consists of seven classes of enzymes, thioredoxins/glutaredoxins reducing system, and low-molecular antioxidant species, which represent reduced and oxidized forms of ascorbate (AsA) and glutathione (GSH). The main classes of AOS enzymes with names of corresponding MetaCyc reactions (*Krieger et al., 2004*) are listed in Table 1. One group of enzymes including superoxide dismutase (SOD), catalase (CAT), glutathione peroxidase (GPX), and ascorbate peroxidase (APX) catalyzes ROS decomposition, and the other one including monodehydroascorbate reductase (MDHAR), dehydroascorbate reductase (DHAR), and glutathione reductase (GR) maintains the level of reduced forms of antioxidants. The AOS components interact with each other by the mechanism summarized in Fig. 1. Superoxides are neutralized in a dismutation reaction catalyzed by SOD, which produces hydrogen peroxide. Hydrogen peroxide can be neutralized in three ways: by CAT, by GPX with the oxidation of reduced thioredoxins, and by APX with AsA oxidation to monodehydroascorbate (MDHA). MDHA can non-enzymatically transform into ascorbate and dehydroascorbate (DHA) and can be restored by MDHAR, which can turn again into ascorbate under the reaction of DHAR with GSH oxidation to GSSG. NADPH-dependent GR can reduce GSSG.

Obtaining of experimental data related to the AOS regulation is the subject of interest for many scientific groups working with different objects. Starting with investigations

**Table 1 Summary of the main classes of the AOS enzymes.** The EC column represents enzyme commission number for reaction, and the Pathway link column addresses to the corresponding reactions in the MetaCyc database.

| Enzyme abbreviation | Full name | EC | Pathway link |
|---|---|---|---|
| APX | Ascorbate peroxidase | 1.11.1.11 | MetaCyc:PWY-6959 |
| GPX | Glutathione peroxidase | 1.11.1.9 | MetaCyc:DETOX1-PWY-1 |
| CAT | Catalase | 1.11.1.6 | MetaCyc:DETOX1-PWY-1 |
| SOD | Superoxide dismutase | 1.15.1.1 | MetaCyc:DETOX1-PWY-1 |
| DHAR | Dehydroascorbate reductase | 1.8.5.1 | MetaCyc:PWY-2261 |
| MDHAR | Monodehydroascorbate reductase | 1.6.5.- | MetaCyc:PWY-2261 |
| GR | Glutathione reductase | 1.8.7.1 | MetaCyc:PWY-4081 |
| NTRC | NADPH-dependent thioredoxin reductase | 1.8.1.9 | MetaCyc:THIOREDOX-PWY |
| – | – | 1.8.4.2 | MetaCyc:GLUT-REDOX-PWY |

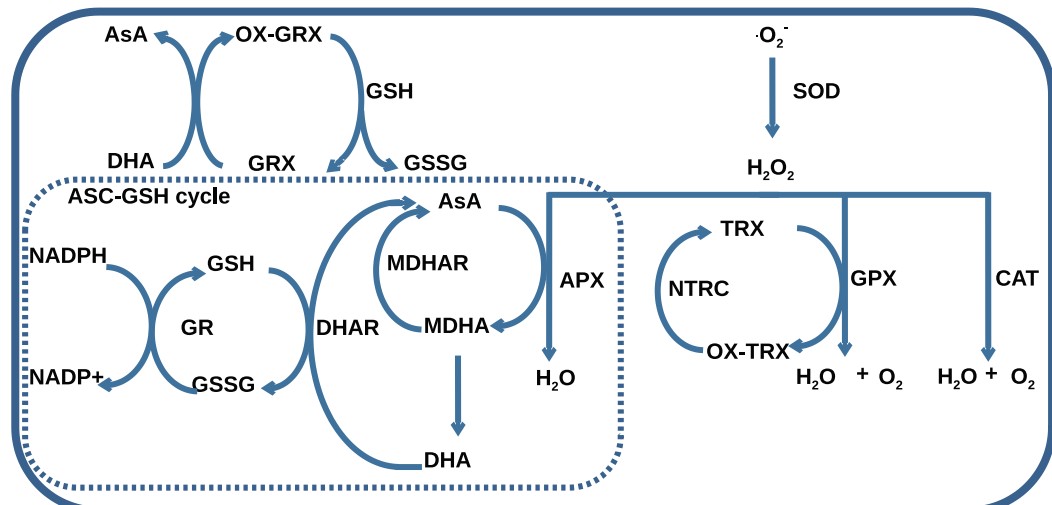

**Figure 1 A general scheme of interaction between AOS components including superoxide dismutase (SOD), catalase (CAT), glutathione peroxidase (GPX), ascorbate peroxidase (APX), dehydroascorbate reductase (DHAR), monohydroascorbate reductase (MDHAR), glutathione reductase (GR), NADPH-dependent thioredoxin reductase (NTRC), ascorbate (AsA), monodehydroascorbate (MDHA), dehydroascorbate (DHA), glutathione (GSH), glutathione disulfide (GSSG), reduced (TRX) and oxidized (OX-TRX) forms of thioredoxin, reduced (GRX) and oxidized (OX-GRX) forms of glutaredoxin.** Ascorbate-glutathione cycle is indicated as "ASC-GSH cycle".

including northern blot analysis/PCR techniques and/or measurements of enzyme activity during stresses, which revealed the ambiguity of the transcription regulation and, in some cases, indirect relationship between changes of mRNA level and enzyme activity during treatment for *Nicotiana plumbaginifolia* L. (*Tsang et al., 1991*), *Raphanus sativus* L. (*Lopez et al., 1996*), *Arabidopsis thaliana* L. (*Kandlbinder et al., 2004*). Recent studies address more precise ways to quantify antioxidant genes using qRT-PCR methods for *Musa acuminata* L. (*Taufikurahman & Widiyanto, 2016*) and *Solanum tuberosum* L. (*El-Argawy & Adss, 2016*) and also showing advanced techniques of choosing the reference

genes for normalization of AOS gene expression for *Cucumis melo* L. (*Kong et al., 2014*) and *Actinidia deliciosa* L. *Petriccione et al. (2015)*.

Also, for many AOS enzymes, the post-transcriptional character of regulation is detected. It was shown that the cytosolic APX, for *Pisum sativum* L., is regulated by control of protein synthesis, probably at the level of initiation of translation during drought (*Mittler & Zilinskas, 1994*). For gene encoding the second catalase subunit, light dependance of mRNA accumulation was found (*Ni & Trelease, 1991*). For another essential AOS enzyme, SOD, it was established that the expression of two Cu/Zn SOD genes regulated by miR398 (*Sunkar, Kapoor & Zhu, 2006*). Expression of the miR398 repressed during stress conditions, which allows the system to activate the targeted SOD genes for protection. Also, an investigation of *Alscher, Erturk & Heath (2002)* showed increased activity in response to the stress of various SODs and a high degree of similarity of their upstream regulatory regions, which indicates the co-regulation of SOD.

The AOS reveals a compensatory behavior: plants with suppressed expression of APX induced the expression of SOD, CAT, GR (*Willekens et al., 1997*; *Rizhsky et al., 2002*). CAT and APX cannot fully compensate each other for their insufficiency, which has been shown for both abiotic and pathogen-caused stresses (*Mittler et al., 1999*). Also, plants lacking APX and CAT demonstrate higher resistance to oxidative stress than mutants defective by only one class of enzyme (*Rizhsky et al., 2002*), which is associated with a rearrangement of plant metabolism and a decrease in photosynthesis activity of double-antisense mutants. Induction of individual enzymes of the AOS can affect various related processes in the cell. For example, *Chen, Twito & Miller (2014)* stressed out the role of ascorbate peroxidase 6 (APX6) in cellular metabolism, including a link between APX6 functioning, ROS signaling, stages of seed development, and up-regulation of abscisic acid (ABA) and auxin for *A. thaliana*.

Accumulated over many years, the existing scientific background and experimental data could serve as a unique basis for the systems biology research of the plant AOS. Therefore we aimed to systematize available data and methods, including mathematical modeling, and to highlight the latest discoveries concerning the plant antioxidant system by describing the underlying biochemical and kinetic properties and subcellular localization.

The primary purpose of this review is to show available data and methods which can help to reconstruct and describe the behavior of the AOS in silico. We focused on the main features of this system, which determines its complexity. We will start by describing the underlying biochemical and kinetic properties of the system. Then we discuss available experimental data and give readers some facts about the regulation of this system during stress. We also emphasize the importance of AOS's subcellular properties, which should be studied more precisely to gain a more profound knowledge of this system. Finally, we discuss available models of the AOS and the mathematical modeling methods related to this system.

## SURVEY METHODOLOGY

Following the review plan, lists of key terms for each research topic were compiled. We searched for articles in the Google Scholar database (https://scholar.google.com/)

using search terms that match the issues of the relevant sections indicated in Table S1. The obtained articles were initially screened based on their abstracts. The selected articles for each section of the review and its quantity are presented in Table S1. This review also pays attention to some biochemical researches which address the study of antioxidant enzyme activities in plants and provide insights into the regulation of the system and its compartmentalization. For searching these articles, we use keywords like kinetic measurements/enzyme activities of the AOS of plants. Because the biochemistry of the antioxidant system was intensively studied in the 1990–2000s, we found it essential to include these articles in the review.

## Main features of the antioxidant system components

The mathematical description of biochemical reactions based on the observed physical phenomena performs a clear understanding of underlying biological complexity. This section assembles the available information found concerning various mathematical descriptions for the AOS components. We provide the kinetic laws proposed by different authors in their original researches. Nevertheless, we admit that later, during the integration of our knowledge about systems components into a single mechanism-based model, some kinetic laws might be reconsidered.

*Ascorbate peroxidase* (APX, EC 1.11.1.11) like an enzyme from 1 heme-peroxidase class, binds hydrogen peroxide to form two intermediate compounds (*Welinder, 1992*) and is inactivated at low concentrations of ascorbate (*Miyake & Asada, 1996*). Hydrogen peroxide-mediated inactivation of an enzyme occurs in the absence of ascorbate (*Kitajima, 2008*). Thus, the kinetics of the enzyme is non-linear and depends on the physiological state of the cell and its redox status.

*Glutathione peroxidase* (GPX, EC 1.11.1.9), unlike APX, does not contain a heme cofactor and can restore itself by two main types of mechanisms: (i) directly reduction by 1-Cys mechanism or (ii) involving thioredoxin by 2-Cys mechanism (*Bela et al., 2015*). The catalytic efficiency (Kcat/Km) of the plant GPX is similar to other peroxidases ($10^3$–$10^6$ M$^{-1}$ s$^{-1}$), but lower than that of APXs, CATs and human GPX ($10^7$–$10^8$ M$^{-1}$ s$^{-1}$) (*Bela et al., 2015*). Even though the role of GPX in the utilization of peroxide for plants is ambiguous in comparison with animals (due to presence of advanced ascorbate system in the cell), nevertheless, it has a similar number of paralogs as APX in plants, and many copies are highly conservative protein structure *Doroshkov & Bobrovskikh (2018)*. The term "glutathione peroxidase" itself is inaccurate for this enzyme since it has been shown that the enzyme is the 5th group of thioredoxin-dependent thiol peroxidases (*Navrot et al., 2006*) and uses thioredoxins as a substrate, and not glutathione. *Herbette, Roeckel-Drevet & Drevet (2007)* also reflected this idea. Therefore, thioredoxin reductases restoring thioredoxins might also be considered as the attendant reducing enzyme for AOS functioning. It is known that thioredoxin systems occur ubiquitously in cell (*Gelhaye et al., 2005*), and thioredoxin reductases have an impact on stress sensitivity (*Serrato et al., 2004*). A detailed discussion of the thioredoxin system is beyond the scope of this review; for reference, see the review of *Balsera & Buchanan (2019)*.

*Catalase* (CAT, EC 1.11.1.6) consists of four subunits, which impose features on its kinetics (*Aronoff, 1965*). In particular, the entire complex's assembly is necessary for the activity of the enzyme, which imposes restrictions on the activity of CAT under conditions of low hydrogen peroxide concentrations. Investigations of CAT kinetic revealed intermediates and saturation point of its reactions with hydrogen peroxide (*Chance & Oshino, 1971*). The mechanism is further analyzed by synthetic mimics of the dimanganese CAT enzymes (*Pessiki & Dismukes, 1994*). The dismutation mechanism is carried out by two consecutive two-electron steps, which are coupled to redox transformation of the two redox centers of catalyst: $Mn_2^{III,III} \rightarrow Mn_2^{II,II}$.

Kinetics of *superoxide dismutase* (SOD, EC 1.15.1.1) reaction can be described as first-order (*Sawada & Yamazaki, 1973*). *Beckman et al. (1992)* demonstrated that the rate of superoxide neutralization by SOD could be limited by the complicated nature of the substrate rather than the concentration of the enzyme, that is, the reaction proceeds without the formation of a long-lived intermediate complex of superoxide and enzyme.

Enzymes that reduce antioxidants have sophisticated kinetic laws. *Glutathione reductase*, GR, EC 1.8.1.7 (*Vanoni et al., 1990*), and *monodehydroascorbate reductase*, MDHAR, EC 1.6.5.4, (*Hossain & Asada, 1985*) are described in terms of "ping-pong" reactions kinetics. *Dehydroascorbate reductase* (DHAR, EC:1.8.5.1) might have a more complex reaction mechanism (*Shimaoka, Miyake & Yokota, 2003*), the so-called "bi-uni-uni-uni-ping-pong" mechanism, in which binding of DHA to the free reduced form of the DHAR was followed by binding of GSH.

For the normal functioning of the AOS, it is essential to maintain the intracellular concentration of low-molecular antioxidants (*ascorbate and glutathione*). Ascorbate is a primal antioxidant of the plant cell (*Njus & Kelley, 1991*). In addition to the fact that ascorbate is a cofactor of the ascorbate peroxidase enzyme, it can directly react with several types of ROS: hydroxyls, singlet oxygen, and superoxides (*Buettner & Jurkiewicz, 1996*). Ascorbate is present in millimolar concentrations in photosynthetic and non-photosynthetic tissues (*Foyer, Rowell & Walker, 1983*). *Chen et al. (2020)* found that adding exogenous ascorbate is increases the regeneration process and induces the development of fruiting bodies in *Hypsizygus marmoreus*.

Glutathione has many functions in the plant cell, among which the most important is antioxidant activity (*Noctor et al., 2012*). Most glutathione is synthesized in chloroplasts and transported to the phloem (*Mendoza-Cózatl et al., 2008*). *Adams et al. (2020)* emphasizes that remediation of glutathione is crucial for alleviation cesium stress in *A. thaliana* and that exogenous application of sulfur-containing compounds increases its efficiency.

In addition to antioxidant enzymes and low-molecular antioxidants, such classes of proteins as *thioredoxins* and *glutaredoxins* play an essential role in the reduction of oxidized forms of antioxidants in the cytosol, nucleus, peroxisomes, mitochondria, and chloroplasts (*Noctor, Reichheld & Foyer, 2018*). Thioredoxins are an evolutionarily ancient family of proteins that have high similarity in the primary structure, but different stability and activity between different evolutionary lines (*Modi et al., 2018*). Peroxiredoxins family plays various roles in different cellular compartments, including antioxidant activity

and signaling (*Lee et al., 2018*); and thioredoxins can serve as electron donors for their reduction.

## Experimental data concerning the antioxidant system components

The development of the kinetic basis and biochemical methods allowed studying the kinetic data and enzyme activities in different degrees of freedom. This section arranges studies addressed to measurements of the total activity of AOS enzymes in substrate-specific reactions.

For plant tissues, the protocol for measuring the activity of APX and guaiacol peroxidase was established by *Amako, Chen & Asada (1994)*. Some studies for tomato showed tissue-organelle and stress-dependent differences in the AOS enzyme activity. For example, *Mazhoudi et al. (1997)* noted that the addition of copper to the soil leads to heavy metal stress for tomato changing the activity of plant antioxidant enzymes; specifically, the activity of APX decreased in stem and whole plants. Investigations of *Mittova et al. (2000)* focused on measurements of antioxidant enzyme activities (APX, MDHAR, DHAR, GR, SOD) in different organelles in leaf and root for two different contrasting tomato species (cv. Lpa is wild and salt-tolerant cv. Lem is cultivated). The authors stressed out that the activity of MDHAR was one order higher than DHAR, which may indicate a dominant role of MDHAR in ascorbate regeneration. Also, salt-tolerant genotype Lpa had a higher SOD/APX ratio in all organelles, which would correspond to its higher tolerance. Their follow-up studies (*Mittova et al., 2002*; *Mittova et al., 2004*) showed different regulation of AOS enzyme activity for plastids, mitochondria, and peroxisomes in roots of Lpa and Lem tomatoes. A study of ROS, AOS activities, and expression patterns for bluegrass (*Bian & Jiang, 2009*) showed that the production of ROS (mainly superoxide) and AOS enzymes activity increased during drought stress. Surprisingly, the expression of most genes that correspond to AOS enzymes was decreased both in leaves and roots. *Taufikurahman & Widiyanto (2016)* studied the overall expression of CAT and APX in control and under different chromium concentration treatments for *Musa acuminata*. The CAT and APX expression levels in roots increase 9 and 3 times under chromium stress of 200 ppm, respectively. However, for shoot tissues, authors revealed weaker upregulation than in roots, which strongly support the tissue-specific and enzyme-specific regulation of the antioxidant response besides the differences in the compartmental organization of the AOS, biochemical features of tissues impact AOS properties. New methods for measurements of antioxidant content are still finding out, for example, ferric-bipyridine assay (*Naji et al., 2020*). The subcellular distribution of ROS was localized by methods of the transmission electron microscope in organelles of leaves (*Zhuang et al., 2019*). Also, for the technique of RT-PCR to expression studies, it is crucial to use the correct reference genes with stable expression under abiotic treatments, as demonstrated for CAT genes of *Cucumis melo* by *Kong et al. (2014)*. The authors revealed that levels of CAT were up-regulated after 3 days of inoculation with *Fusarium wilt* in roots, and were significantly overestimated when using the wrong reference gene for normalization. In addition, various methods for calculating the stability of reference genes and using them in combination is fruitful for accurate quantification of

the expression levels of antioxidant genes (*Petriccione et al., 2015*). Unfortunately, nowadays, there are no precise tools for counting ROS in cellular compartments.

Recent studies describe the stress reaction of the AOS system more comprehensively considering other biological aspects and great importance of this system to stress response. *Hippler et al. (2016)* established the role of citrus rootstocks in the up-regulation of antioxidant enzymes (Cu/Zn SOD, CAT, APX) in leaves and nutritional status during long-time copper treatment. Additional investigations on two citrus genotypes showed a direct relationship between an increased antioxidant activity and stress tolerance (*Zandalinas et al., 2017*). Also, there is evidence of a connection between the expression of plastid antioxidant genes and cold-acclimations (*Juszczak et al., 2016*). Investigation of drought stress effect to photosystems and AOS (*Guo et al., 2018*) revealed that photosystem I (PSI) is more sensitive to stress than photosystem II (PSII) and process of photoinhibition related to ROS accumulation. Also, drought-resistant plants of sorghum genotype M81E had a significantly higher activity of APX and SOD during stress. *Bhuyan et al. (2019)* studied AOS components and glyoxalase systems for different wheat genotypes in order to detect its resistance. Study of the effect of salinity on tomato (*Parvin et al., 2019*) showed a strong increase of SOD, APX, GPX, and DHAR activity proportionally to NaCl concentration during salt stress. *Riffat & Ahmad (2020)* showed for *Zea mays* L. that under salt-stress treatment, amounts of ascorbate and tocopherols decreased and production of SOD, CAT, POX, and some compounds (i.e., phenolics, carotenoids, MDA) increased. The authors emphasized the role of exogenous sulfur as an inductor of salt tolerance because of enhancement of SOD, POX, CAT, increasing of AsA, tocopherol, phenolics, and decreasing carotenoids and MDA during sulfur supply. The large-scale meta-transcriptome and complementary experimental analysis of antioxidant genes regulation in response to cold and water deficiency for rice and bread wheat were carried out (*Ermakov et al., 2019*). As a result, patterns of regulation of different gene copies of AOS in response to stress were described, and cold response turned out to be more stable and homogeneous than salinity, perhaps due to peculiarities of used methods for stress conditions.

Several studies were devoted to estimations for the quantification of antioxidants. Measurements of dynamical concentrations of ascorbate and glutathione in response to ozone exposure (*Castillo & Greppin, 1988*) showed that the total amount of antioxidants is maintained at approximately the same level, even so, under stress exposure, the number of oxidized forms and the activity of DHAR restoring ascorbate increases. Glutathione concentrations are approximately 1–3.5 mmol in chloroplasts and 0.1–0.7 mmol in other compartments (*Rennenberg, 1980*). The highest concentrations of AsA were detected in cytosol and nuclei (*Zechmann, Stumpe & Mauch, 2011*), while levels of AsA in plastids and mitochondria were about half of cytosolic AsA. However, the most definite increase of ascorbate was detected in the vacuole (fourfold) and chloroplasts (twofold). It was shown that concentrations of the thylakoid APX and Cu-Zn SOD reach 1 mmol (*Asada, 2006*) nearby of PSI in the chloroplast, but stromal APX has a much lower ratio of concentration. Generation ratios of hydrogen peroxide vary by three orders between organelles

(*Foyer & Noctor, 2003*) from 100 in mitochondria to 10,000 nmol m$^{-2}$ leaf surface s$^{-1}$ in the peroxisome.

Methods of metabolomics also might give a better understanding of the processes related to AOS. For instance, increased amounts of phenolic acids and flavonoids detected on 120 h germination stage compared with the 36-h stage (*Chu et al., 2020*). Also, the publicly available data of gene expression levels, such as Gene Expression Omnibus (*Clough & Barrett, 2016*), Expression Atlas (*Petryszak et al., 2016*) can help to analyze dynamics of the AOS.

All this heterogeneous data can help to understand the complex properties of AOS in normal conditions (ratio of reduced/oxidized forms of antioxidants, units of enzyme activity, expression levels of individual genes, generation of ROS). Still, it seems that the stress response of the AOS is much more complicated and includes changing the activity of AOS enzymes, amounts of antioxidants, and unambiguous transcriptional regulation of the coding genes of antioxidant enzymes either cellular ROS and antioxidant interplay. Also, it seems like the chloroplastic antioxidant system has a unique role in whole-cell ROS metabolism. For example, chloroplasts have the highest hydrogen peroxide generation (*Foyer & Noctor, 2003*), and strong increasing of AsA concentrations under high-light stress (*Zechmann, Stumpe & Mauch, 2011*). Besides that, under high-light stress conditions, modulation of the chloroplast antioxidant system by exogenous spermidine is enhanced tolerance to salinity-alkalinity stress for tomato (*Li et al., 2015*).

## Stress-response of the antioxidant system components

The complex nature of the AOS results in diversified regulating inductors of this system on multiscale levels. ROS signaling pathways are involved in systemic acquired acclimation to stresses, and in a non-specific way they may be integrated into other acclimation pathways (*Gilroy et al., 2016*). Hormones, such as salicylic acid (*Wang et al., 2015b*), brassinosteroids (*Vardhini & Anjum, 2015*), and combination of the latter with kinetin (*Ahanger et al., 2020*) enhance the activity of the AOS enzymes and low-molecular antioxidants. A similar effect is also observed for ethylenediaminetetraacetic acid (EDTA) (*Habiba et al., 2015*). It was also found that some of the organic chemicals, that is, ferulic acid (*Yildiztugay et al., 2019*), and trehalose (*Mostofa, Hossain & Fujita, 2015*) induce the AOS response to stresses (boron and salt, respectively). Alpha-lipoic acid (*Terzi et al., 2018*) had a similar effect: the activity of SOD, CAT, GPX, GR, and MDHAR increased in osmotic stress but decreased the activity of the APX. Treatment with important inorganic substances, such as calcium (*Rahman et al., 2016*), silicon (*Kim et al., 2017*), sulfur (*Adams et al., 2020*) and selenium (*Alyemeni et al., 2018*) also can enhance AOS characteristics. Exogenous hydrogen peroxide (*Guler & Pehlivan, 2016*) can improve drought tolerance by enhancing AOS activity. This fact suggests that there is a cross-talk in the regulation of the system to different inductors.

The AOS has a versatile regulation during abiotic and biotic stresses. *Abiotic stress* is characterized mainly by the increasing activity of antioxidant enzymes and reducing the growing number of radicals, up to depletion of the system's capacity. Abiotic stresses usually induce the AOS to enhance the work on the fight against radicals. The antioxidant

enzymes (APX, SOD, GPX, GR, CAT) up-regulates during various abiotic stresses (*Ahmad et al., 2010*). The other two enzymes, DHAR (*Yoshida et al., 2006*) and MDHAR (*Eltayeb et al., 2007*), also have essential redox functions to protect against abiotic stress. These facts argue in favor of the co-regulation of enzymes of this system in response to stress. Different types of abiotic stress have a common pathway between them. *Roychoudhury, Paul & Basu (2013)* noted that regulation of abiotic stresses had an ABA-dependent (for drought and salinity) and ABA-independent pathways (drought, salinity, cold). Also, transgenic plant lines with additional copies of AOS genes showed higher resistance to abiotic stresses. For example, it was shown on peanut and apple during drought and salt treatment (*Negi et al., 2015*; *Sun et al., 2018*), on cassava in cooling stress conditions (*Xu et al., 2014*). On the other hand, plant lines with knockouts of individual antioxidant genes are less tolerant of stress (*Wu et al., 2015*).

Regulation of the AOS during *biotic stresses* has a common dual pattern. The first pattern involves the activation of the AOS enzymes to protect the cell from ROS. Second, so-called oxidative burst, characterized by a decrease in the antioxidant activity of enzymes and intercellular signaling on the destruction of infected cells. For example, tomato plants infected by *Botrytis sineria* Pers. demonstrated increased activities of SOD, CAT, and GPX during an early stage of treatment. Still, during the advanced stage of disease activity of the AOS enzymes decreased that may cause programed cell death (*Kuźniak & Skłodowska, 2005*). *El-Argawy & Adss (2016)* showed that bacterial treatment (*Ralstonia solanacearum*) causes a genotype-specific changing expression of peroxidases, polyphenol oxidase, and CAT on different varieties of *Solanum tuberosum*. Interestingly, polyphenol oxidase levels, especially peroxidase, were higher in highly resistant cv. Nicola and medium resistant cv. Kara. Vice versa, the CAT level was highest for susceptible cv. Spunta and lowest for cv. Nicola. It may indicate a more successful oxidative burst strategy and programed cell death with bacterium-resistant genotypes with reduced CAT levels. During the oxidative burst, NADPH oxidases generate a large amount of ROS, which acts as signaling molecules for apoptosis and can trigger cell death programs to prevent the spread of infection throughout the plant (*Torres, 2010*). A comprehensive scheme of the signal role of ROS in pathogen interactions described by *Camejo, Guzmán-Cedeño & Moreno (2016)*.

The question about the effects of *stress combination* is still open. A comprehensive review by *Suzuki et al. (2014)* focused on interactions of abiotic and biotic stresses. The authors revealed unique response patterns of stress combinations that can't be predicted by the special effects of stress. *Atkinson, Lilley & Urwin (2013)* investigated the transcriptome response of *A. thaliana* on drought stress, pathogen infection, and summarizing the result of stress combination using microchip technology. During this study, the authors revealed about 50 multi-stress-regulated genes with a unique pattern of regulation. Also, another study on the *A. thaliana* transcriptome in response to drought and pathogens individually and in combination (*Davila Olivas et al., 2016*) showed that clustering of stress combinations was by type of stress in case of early response (different types of stress are clustered separately) and by time in case of late response (different stresses with the same duration are clustered in the same groups). These facts may indicate a higher specificity of regulation of the antioxidant system to stress in the

early response than in the late response. *Atkinson & Urwin (2012)* gave numerous examples showing the breeding of varieties resistant to abiotic stresses can make hybrids more vulnerable to pathogens, and vice versa. Thus, strategies for protecting against abiotic and biotic stresses are different and often opposite. However, common property for abiotic, biotic stress, and development processes is their signaling system, directly related to NADPH oxidase activity and an increase in ROS production (*Suzuki et al., 2011*). Thus, a delicate balance between the number of radicals and the activity of the components of the antioxidant system occupies a central place in the study of such a complex issue of the interaction of ROS, AOS, and cell homeostasis.

In the AOS, there are individual enzyme-coding genes that have high importance for the functioning system in general, especially in stress conditions. *Doroshkov & Bobrovskikh (2018)* found the relationship between the evolutionary features of antioxidant enzymes and their expression levels for predicting prospective targets for selection across cereals. The authors identified groups of highly conserved genes with a high level of mRNA expression and a group of less conservative with a low-level expression. However, a group of highly conserved genes was isolated, but the expression under normal conditions was low. These genes may be activated in response to specific effects, such as stress. This method seems to be a prospective approach for isolating general genetic components of this system. The lack of information about the complex regulation of the system in response to stressful effects makes it difficult to identify specific trends and patterns of its behavior. It is mostly because the AOS behavior is associated with multi-copy of system components, subcellular compartmentalization, and a considerable number of freedom degrees in its regulation, including different stages of plant development, specificity of photosynthetic and non-photosynthetic tissues and various types of stress.

## Subcellular compartmentalization of the antioxidant system components

The plant cell contains several compartments depending on its type and/or part of a particular tissue, and each compartment has its own source of ROS. The compartmentalization of a plant cell determines the specifics of the metabolism and the set of metabolic reactions that can occur, allowing the different compartments of the cell to perform unique functions. The AOS also is divided into a set of subsystems functioning in different compartments and tissues. In the various cells, compartments differ in the set of enzyme classes of the AOS (*Mittler et al., 2004*), conditions (pH, metabolite composition, the ratio of reduced and oxidized antioxidants) and various ratios of the activities of individual isoforms of antioxidant enzymes. Also, it is known that ROS generation rates vary widely between compartments (*Foyer & Noctor, 2003*). Thus, different organelles have various abilities to neutralize radicals. Herein, we proceed to consideration of the different cell compartments in the context of the generation of radicals and systems to combat them. Based on the researches (*Palma et al., 1991*; *Foyer & Noctor, 2003*; *Rhoads et al., 2006*; *Cho et al., 2009*; *Suzuki et al., 2011*; *Jarvis & López-Juez, 2013*) reviewed in this article we propose the following structural model of the interplay between AOS subsystems and components and the underlying metabolic processes producing ROS under oxidative stress conditions

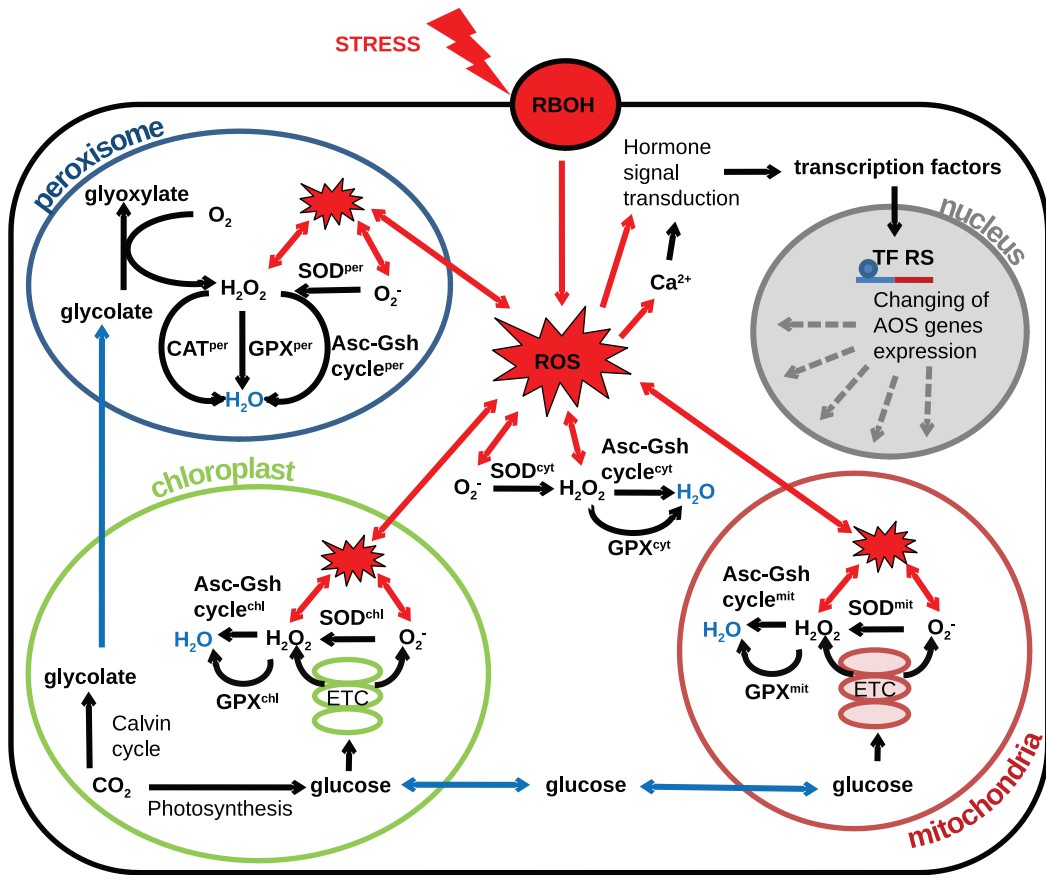

**Figure 2 Revised structural model of the interplay between the AOS components (black arrows) and the basic metabolic processes producing ROS (blue arrows correspond to transport) under oxidative stress conditions (red arrows correspond to stress-induced processes).** The generalized plant cell is represented as membrane and cytosol (superscript "cyt") containing the following compartments: mitochondria (superscript "mit"), chloroplast (superscript "chl"), peroxisome (superscript "per"), and nucleus. Components of the AOS with a superscript corresponding to the cell compartment stand for a set of paralogues enzymes localized in this compartment. The following abbreviations are used in the figure: respiratory burst oxidase homologs (RBOH), reactive oxygen species (ROS), superoxide dismutase (SOD), catalase (CAT), glutathione peroxidase (GPX), ascorbate-glutathione cycle (Asc-Gsh cycle), electron transport chain (ETC), transcription factor regulatory sequence (TF RS).

Fig. 2. In the model, the generalized plant cell is represented as a membrane and cytosol containing the following compartments: mitochondria, chloroplast, peroxisome, and nucleus. Features of the AOS functioning in these compartments are described below.

*Chloroplasts* are the main source of ROS in a plant cell (*Foyer & Noctor, 2003*) and essential organelle for generating energy. The main processes which generate ROS as byproducts in chloroplasts are associated with photosynthesis and activity of the electron-transport chain complexes. The activity of photosystems during the day varies depending on the intensity of sunlight. Due to the special conditions of chloroplast functioning (including variable illumination and electron flow), chloroplasts are required to flexible regulation of metabolic processes. Such versatile control was demonstrated by *Jarvis & López-Juez (2013)* for the regulation of the electron balance between

photosystems. In particular, the redox status of an important component of the electron transport chain, plastoquinone, is controlled by activating the STN7 and CSK gene cascades that affect the expression of genes to alter the electron balance in photosystems. The signal molecule in this process is oxygen, which is produced in high light conditions. Besides, oxygen produced under high light conditions is the signal molecules in this process. Oxygen can induce the expression of genes involving the stress response pathway (mainly antioxidant genes), or cause cell death. In the absence of the plastid EX1 and EX2 genes, the accumulation of oxygen not causes changes in gene expression and the implementation of the cell death program. Hydrogen peroxide, which is generated by chloroplasts under high light conditions, triggers the transcription of APX2 (cytosolic APX) and other genes in vascular cells, usually through an abscisic acid-induced pathway (*Galvez-Valdivieso et al., 2009*).

Also, chloroplasts have NADPH-thioredoxin reductase (NTRC) (*Serrato et al., 2004*), which can restore the disulfide bridges of target proteins by using NADPH. Therefore, the NTR/TRX system plays a vital role in chloroplasts and allows maintaining the redox status of the compartment. In chloroplasts, there is Cu-Zn SOD, which transfers superoxides into hydrogen peroxides. The ascorbate-glutathione cycle (*Lidon & Henriques, 1993*) neutralizes hydrogen peroxides. APX neutralizes hydrogen peroxide, using ascorbate as an electron acceptor. Approximately 50% of APX is found directly in the thylakoids (*Polle, 1997*). DHAR and GR mediate the reduction of oxidized ascorbate. Since the rate of non-enzymatic reduction of dehydroascorbate by reduced glutathione (GSH) is relatively high (*Polle, 1997*), enzymatic reduction in chloroplasts appears to be of secondary importance. Unlike other antioxidant enzymes, GR appears to be directly regulated by the redox status, partially inhibited by NADPH, and activated by hydrogen peroxide and oxidized glutathione (GSSG) (*Polle, 1997*).

The main feature of AOS in *peroxisome* is the presence of catalase, which plays a significant role in maintaining redox status. For example, transgenic catalase-deficient leaves of tobacco had necrotic lesions during intensive light conditions (*Willekens et al., 1997*), but not in low-light conditions. Also, these transgenic plants in high-light conditions accumulated a lot of oxidized glutathione and reduced the amount of ascorbate by four times. Plants with defective catalase accumulate hydrogen peroxide (*Hu et al., 2010*) and have many adverse physiological effects, such as growth suppression and reduced photorespiratory activity (*Bueso et al., 2007*; *Queval et al., 2007*). However, works are proving that the ascorbate-glutathione cycle with its components plays an essential role in the peroxisomal fight against ROS. Thus, it has been shown by *Ribeiro et al. (2017)* that knockout plants by peroxisomal APX show signs of oxidative stress and early aging, while overexpression is associated with increased plant protection against stress (*Wang, Zhang & Allen, 1999*). Peroxisomes perform multiple catabolic processes of many substances (fatty acids, amino acids, polyamines, etc.), which generate ROS inside them. Also, most plants are characterized by C3 photosynthesis, which forming glycolate. Then glycolate transported from chloroplasts to peroxisomes, where it is oxidized to glyoxylate (*Kisaki & Tolbert, 1969*). The byproduct of this reaction is hydrogen peroxide,
which is released in significant quantities (*Foyer & Noctor, 2003*). So, it is essential to maintain a balance between the production and scavenging of ROS in peroxisomes, which clearly illustrated by *Wang et al. (2015a)*. Authors showed that redox status regulates peroxisome biogenesis and degradation what plays a role in the maintenance of integral multi-compartmental cell metabolism. A distinctive feature of peroxisomes is their high plasticity in response to various effects. For example, *Rodrguez-Serrano et al. (2009)* indicate that peroxisomes had significantly increased dynamics and speed of movement during the treatment with $CdCl_2$. It has been shown that the number of peroxisomes increases during clofibrate treatment (*Palma et al., 1991*). However, during oxidative stress, the biogenesis of peroxisomes can be repressed, which may be connected with a violation of the peroxisomal protein importers. This hypothesis is supported by evidence that several peroxins, proteins involved in peroxisome biogenesis, are known to be modified or violated by an imbalance of cellular ROS (*Ma et al., 2013*).

*Mitochondria* provide the energy for the plant cell and regulate overall metabolism through respiration. The main reactions in this organelle are phosphorylation of ADP and oxidation of various substrates that generate ROS. Review (*Noctor, De Paepe & Foyer, 2007*) considers the integrative role of mitochondria in the generation of radicals. Due to photosynthetic processes in the plant cell, mitochondria exist in an environment enriched with oxygen and carbohydrates. Mitochondria regulate their redox homeostasis by regulating the flow of electrons through the electron transfer chain and the change in the activity of NADPH-dependent oxidases. However, the absolute levels of radical production during mitochondrial metabolism are relatively small (*Foyer & Noctor, 2003*). The main sites of superoxide production in mitochondria are complexes I and complexes III in the ETC (*Møller, 2001*; *Rhoads et al., 2006*). In mitochondria, superoxides are fixed by manganese SOD, and peroxides are eliminated in the ascorbate-glutathione cycle. The investigation of *Fernie, Carrari & Sweetlove (2004)* reveals that the activity of the electron transport chain of mitochondria is associated with the activity of the tricarboxylic acid cycle oxidative phosphorylation, the biosynthesis of carboxylic acids and photorespiration.

Hydrogen peroxide (*Bolouri-Moghaddam et al., 2010*) can easily transport into the *vacuole* from the cytoplasm. It is also possible that the vacuole is capable of generating ROS using its NADPH oxidase, based on proteomic analysis of the tonoplast membrane (*Shi, Wang & Wei, 2006*; *Whiteman et al., 2008*; *Pradedova, Isheeva & Salyaev, 2011*). It is shown that there are peroxidases of class 3, which eliminate ROS and control the level of hydrogen peroxide (*Costa et al., 2008*; *Fini et al., 2011*). However, the non-enzymatic way of utilizing radicals in vacuoles, which includes low-molecular species like flavonoids and antioxidants, seems to be the most important. Vacuoles contain approximately 2 mM of ascorbate (*Zechmann, 2018*) and approximately 100 μM of glutathione (*Pradedova et al., 2018*).

Thereby the AOS is the multi-level system, and it has plant-specific patterns of behavior under various stress conditions (*Shalata & Tal, 1998*; *Gossett, Millhollon & Lucas, 1994*) and in various cellular compartments (*Mittler et al., 2004*).

## Modeling approaches to study the antioxidant system

Although the first models of metabolic pathways of photosynthesis in chloroplasts were developed more than 30 years ago (*Hahn, 1987*; *Laisk & Walker, 1986*), just a few of them describing the components of the AOS in plants. These models refer to various phenomena related to ROS metabolic pathways, with different levels of accuracy.

In the early 2000s, *Polle (2001)* built the first model for the functioning of the ascorbate-glutathione cycle in chloroplasts. This model described the functioning of redox reactions of this cycle with the possibility of calculating the steady-state of the system components. However, the model implied many assumptions. For example, the model includes a constant NADPH concentration, which leads to the accumulation of NADP+ in chloroplasts. Also, the model used a constant rate of formation of $O_2^-$ and the non-specific kinetic mechanisms of reactions for enzymes of different classes (except SOD).

Later, *Valero et al. (2009)* improved this model. In their model, real kinetic mechanisms for various enzymatic reactions were used. The model simulates the conditions of oxidative stress and calculates a steady-state for the AOS components. The innovation of this model is the inclusion of a source of electrons, which is competitively distributed between three competitive routes (radical generation, photoreduction of NADP+ to NADPH, photoreduction of monodehydroascorbate to ascorbate). The simulation established a high sensitivity of the system to the number of substances that restore ascorbate and to the NADPH consumption by the Calvin cycle. In stress conditions, the model reflected the depletion of the system's antioxidant capacity in the sequence of NADPH-glutathione-ascorbate and their reduction in the opposite order. Further, inactivation during the stress effects of GR, APX, and SOD resulted in irreversible inactivation of APX and excessive accumulation of hydrogen peroxide.

Further development of these ideas by the same scientific group (*Valero et al., 2015*) describes the most up-to-date and advanced mathematical model of the AOS. The model studies the dynamic behavior of the ascorbate-glutathione cycle in the dark and light phases of photosynthesis. The system is described by a competitive flow of electrons in three different routes; the choice of the route depends on the intensity of sunlight during the photoperiod (in the dark phase, the electron flux is equal to zero). Unlike previous models provided only stationary solutions, this model allows simulating daily fluctuations of metabolites, electron fluxes, and enzyme activity in an ascorbate-glutathione cycle. The simulation results reflect the importance of the distribution of electron fluxes through the system to deal efficiently with the radicals in the ascorbate-glutathione cycle. The authors noted that their model could help to isolate the most important enzymes and metabolites in this pathway. This approach can also describe strategies for analyzing the mechanisms of plant protection from radicals and can be extrapolated to other metabolic pathways in the future.

When constructing a mathematical model for the AOS, we should pay special attention to the pathways of ROS generation and ROS-associated signaling and various mechanisms of cell response to oxidative stress and ways of adaptation. The system complexity is so high that it might be challenging to keep in mind all interactions and various factors

affecting those interactions. Thus, system biology approaches can help with an overall description of the system. One can describe biochemical interactions in terms of differential equations. For example, every single biochemical reaction can be described with a rate equation based on a specific kinetic law (Michaelis-Menten law, Hill-type kinetics, "ping-pong" mechanisms, etc.). Then the system of differential equations should be solved numerically. Thus, the emergent behavior of a whole system might be reconstructed in silico on the base of knowledge about interactions between systems components.

## CONCLUSION

During the last decades, an extensive compendium of knowledge was accumulated. Still, most of it is fragmentary and concerns various individual properties of the system in control and stress conditions, different compartments (specific features of regulation and sources of electrons), different tissues (tissue-specific AOS profiles), and plant development stages. However, the dynamics and regulation of this system have not been exhaustively studied. In particular, some information concerning intracellular crosstalk between various components of the system in response to stress is missing, and there are still some gaps related to regulation and functional diversity attributed to different intracellular compartments.

Nevertheless, the development of biochemical and cell biology approaches provide much information concerning the functioning of the AOS. For example, the amount of transcriptome data for an extensive compendium of plant species is continuously growing. Now it is a time for integration all these data into a detailed mechanism-based mathematical model that would become an in silico replica of the underlying biology. Currently available data, including biochemical studies of AOS and recent omics data related to this system in combination with in silico methods for simulation and analyzing such systems, can help to make steps towards this long-term goal.

We should admit that there might be various authors' attempts to build different versions of models, for different fragments of the AOS, and different plants. The approach of blueprint modeling could help to deal with this complexity. Blueprint modeling is based o the concept that all organisms are similar in their macromolecular networks. They use similar building blocks (mRNA, DNA, proteins, etc.) and similar biochemical reactions organized in similar metabolism pathways. The difference is mostly in the values of the parameters, for example, in the expression levels of certain enzymes. Thus, a generic blueprint model can be built and then parameterized for every particular instantiation, for example, for every specific organism or condition (*Kolodkin, 2017*).

The ideal way would be to build such a blueprint model as a live online system in the environment, similar to the one proposed by JWS Online (https://jjj.bio.vu.nl/). The models on JWS Online are aligned with FAIR (Findable, Accessible, Interoperable, Reusable) principles and can run online for either default parameters (curated and approved JWS team), or for parameters chosen by the user. The user can choose parameters based on his/her own experimental data or from curated databases, such as

SABIO-RK (http://www.sabio.h-its.org/), or the data assembled at facilities of ELIXIR Research Infrastructure (https://elixir-europe.org/). Now user should do it manually. However, it is planned to do it automatically. For example, the user will go in the database, choose the appropriate organism and conditions, and run model simulations in the JWS Online environment. This is planned to implement within the framework of the recently started EOSC-Life project (European Open Science Cloud) supported by Horizon 2020.

The reconstruction of the emergent behavior of the AOS in silico would help to identify network targets for intervening antioxidant defense, for example, in crop-related biotechnologies. On the other hand, building a systems biology model would be a valuable exercise in itself. The model building would be concise the efforts of many researchers into a single framework and would merge sometimes contradictory and fragmentary information into a unique global picture. This would help to identify possible gaps in our knowledge, would shed light on many blind spots in our understanding of the antioxidant defense, and would drag the experimental research towards new horizons.

## ABBREVIATIONS

| | |
|---|---|
| **AOS** | antioxidant system |
| **ROS** | reactive oxygen species |
| **ABA** | abscisic acid |
| **SOD** | superoxide dismutase |
| **CAT** | catalase |
| **GPX** | glutathione peroxidase |
| **Asc-Gsh** | cycle ascorbate-glutathione cycle |
| **APX** | ascorbate peroxidase |
| **DHAR** | dehydroascorbate reductase |
| **MDHAR** | monodehydroascorbate reductase |
| **GR** | glutathione reductase |
| **AsA** | ascorbate |
| **MDHA** | monodehydroascorbate |
| **DHA** | dehydroascorbate |
| **GSH** | glutathione |
| **GSSG** | glutathione disulfide |
| **POX** | peroxidase |
| **MDA** | malondialdehyde |
| **TRX** | thioredoxin |
| **GRX** | glutaredoxin |
| **NADPH** | nicotinamide adenine dinucleotide phosphate |
| **NTRC** | NADPH-dependent thioredoxin reductase |
| **ETC** | electron transport chain |
| **PSI/II** | photosystem I and II |

### Funding

The study was carried out with a grant from the Russian Science Foundation (Project No. 19-74-10037). The funders had no role in study design, data collection and analysis, decision to publish, or preparation of the manuscript.

### Grant Disclosures

The following grant information was disclosed by the authors:
Russian Science Foundation: 19-74-10037.

### Competing Interests

The authors declare that they have no competing interests.

### Author Contributions

- Aleksandr Bobrovskikh performed the experiments, analyzed the data, prepared figures and/or tables, authored or reviewed drafts of the paper, and approved the final draft.
- Ulyana Zubairova analyzed the data, authored or reviewed drafts of the paper, and approved the final draft.
- Alexey Kolodkin conceived and designed the experiments, authored or reviewed drafts of the paper, and approved the final draft.
- Alexey Doroshkov conceived and designed the experiments, performed the experiments, analyzed the data, authored or reviewed drafts of the paper, and approved the final draft.

### Data Availability

This review accumulates exclusively literary sources and does not contain original raw data or code.

### Supplemental Information

Supplemental information for this article can be found online at http://dx.doi.org/10.7717/peerj.9451#supplemental-information.

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
