# Peer review of "Subcellular compartmentalization of the plant antioxidant system: an integrated overview"

_PeerJ, doi:10.7717/peerj.9451_

## Round 0.1 · original submission · Minor Revisions

Please take into consideration the reviewer's comments, and provide a revised manuscript and a detailed point-by-point rebuttal letter.

Reviewer 1 ·

Basic reporting

The manuscript is well written, professional English was used throughout the manuscript.

Basically the paper put together a revision of the plant antioxidant system in an integrative way. The main concepts are correctly explained according to up to date understanding of antioxidant enzyme families and antioxidant molecules. The major point is to look and integrate knowledge about this system at different cell locations, like the cytosol, mitochondria, chloroplast and peroxisomes.

In line 310, the name of the fungi is misspelled, is it Botrytis cinerea Pers.?

In line 323, the word arabidopsis should be Arabidopsis

In line 368, it says in chloroplasts associated, I suggest: .. in chloroplasts are associated ...

In line 441, do you mean ascorbate and glutathione are present or presented in vacuoles?

Experimental design

The method to select experimental work published through the years is clearly described after the Introduction.

Validity of the findings

Novel analysis of available information about the antioxidant system is presented. As part of this analysis the discussion is reasonable and authors synthesized the information in an organized way. Although the analysis was kind of historical, the length is appropiate to get a sense of how this field has advanced. It also, authors add insight to what could be done ahead to make more integrative studies.

I consider important the way they uncover current opportunities to set up enzymatic assays that are still lacking.

Additional comments

My suggestion is about information at Table 1, I am not sure about the relevance of the pathway link column, perhaps if this part is explained in the text it will help to understand how it can be helpful.

Reviewer 2 ·

Basic reporting

The paper is well written and the english language is good

Experimental design

no comment

Validity of the findings

Authors described their methodology and made a really good work of synthesis in the AOS components part. This paper also shed the light on modeling approaches to study antioxidant system and this is a very interesting part.

Additional comments

I only have a minor suggestion/modification: I was wondering why thioredoxins and glutaredoxins did not appear in the introduction part, in abbreviations and in table 1. First I was thinking that authors have forgotten these components since they talk about them in the main features of the antioxidant system components. Maybe authors could consider thioredoxins and glutaredoxins as the other components, because their choice of presnetation it is a little disturbing, as if thioredoxins and glutaredoxins were of less importance.

·

Basic reporting

This is a very interesting review which summarizes much of the scientific knowledge about antioxidants systems in plants, putting together aspects of biochemistry, plants biology and subcellular integration.
The review is correctly organized in sections including different fields of the Antioxidant System (components, regulation, compartmentalization, modelling).). It is appreciable that a large bibliographic research was done, including both up-to-date and also some outdated bibliography
English grammar requires extensive editing. Authors should pay careful attention to some phrases or sentences which are confusing.

Experimental design

The survey methodology is very simple and concise and it is well described.
Bibliography is very abundant, covering all the aspects of the topics reviewed.
Some minor mistakes were found such as sources missing in the bibliography section and some citations incorrectly written.

Validity of the findings

no comment

Additional comments

Some suggestions were included in the attached pdf which, in my opinion, are important for the understanding of this review.

---

## Round 0.2 · accepted · Accept

The manuscript has improved over the review rounds and it is now accepted at PeerJ.